# Quantitative Analysis of Online Labor Platforms’ Algorithmic Management Influence on Psychological Health of Workers

**DOI:** 10.3390/ijerph20054519

**Published:** 2023-03-03

**Authors:** Gengxin Sun

**Affiliations:** College of Computer Science & Technology, Qingdao University, Qingdao 266071, China; sungengxin@qdu.edu.cn

**Keywords:** public health, online labor platforms, algorithmic management, psychological health, take-out riders

## Abstract

Online labor platforms (OLPs) can use algorithms to strengthen the control of the labor process. In fact, they construct work circumstances with higher work requirements and pressure. Workers’ autonomy in behavior is limited, which will have a great influence on their labor psychology. In this paper, taking the online take-out platform as an example and by using a qualitative study of take-out riders’ delivery processes, which were supplemented by semi-structured, in-depth interviews with online platform executives and engineers, we used grounded theory to explore the influencing factors of OLPs’ algorithmic management on take-out riders’ working psychology. The quantitative analysis results showed that, in the context of conflict between work autonomy and algorithmic management, platform workers experienced psychological tensions relating to work satisfaction, compensation, and belonging. Our research contributes to protect public health and labor rights of OLP workers.

## 1. Introduction

With the deep application of the Internet and algorithm technology in the field of labor, the gig economy expands rapidly. In the gig economy, the construction of internet platforms and the application of algorithmic management enable employers to use online labor platforms (OLPs) to achieve direct and comprehensive contact with freelancers, so as to complete work tasks assignment of different complexities. The OLPs are attracting a large number of independent contractors and freelancers in a flexible way. In Europe and the U.S., 20–30% of the working-age population perform temporary, flexible jobs in OLPs [1]. In China, the number of participants in the gig economy had been about 830 million in 2020, including about 84 million OLP workers [2]. OLPs are considered as a new employment mode. It is believed to give workers the power to allocate time and energy independently through information technology [3,4]. Yet, OLPs also offer a mode for organization of work, which relies on the use of algorithms to monitor and control platform workers. Therefore, under the working circumstances of real-time monitoring algorithm technology, the OLP workers’ autonomy seems to be more limited. At the same time, workers work hard under the control of real-time quantitative and hidden intelligent platform algorithms, which not only leads to increasing workload, but also affects the physical and mental health of workers. Sun et al. [5] point out that take-out platforms increase the control and predictability of meal delivery through giving consumers a kind of “God’s Vision” that overlooks the overall situation. This vision adds considerable invisible mental pressure to take-out riders, and the rider’s flexibility in delivering meals is greatly compromised. Based on the power dependence theory, Feng et al. [6] concluded that, in online labor platforms, due to the influence of the power dependence relationship, the reduction of autonomy further leads to over-investment in working time and physical exhaustion, thus affecting the health of workers, and workers seem unable to control when to stop working, thus falling into a negative cycle of work engagement upgrading.

According to the definition of the World Health Organization, people with positive mental health are full of positive emotions and can work effectively. Relevant research [7] shows that the overall performance of employees with positive mental health is 16% higher than that of colleagues with negative mental health. Due to the increasing workload and work pressure of workers on the online labor platforms, and the limited space for career development, workers not only easily to lose their enthusiasm for work, but also think that their work is worthless. Therefore, the research on the potential influence of OLPs on workers’ psychology is of great significance to comprehensively grasp the influence of OLPs’ algorithmic management on work autonomy and labor control.

The take-out industry is the fastest growing field in the gig economy; the number of take-out riders in China is close to 7 million. Take-out platforms have become the most typical and representative OLPs. With the emergence of a large number of take-out riders, urban problems, such as increasing traffic violations and traffic accidents, have also taken place [8]. These problems are caused by the shorter and shorter delivery time, strict requirements from customers, and severe punishment system, which also reflect the algorithmic management labor control of OLPs over take-out riders and the resulting psychological pressure. Relevant research [9,10] found that there was uncertainty in delivery time and customer evaluation in delivery labor. Take-out riders developed emotional labor strategies for different subjects in the labor process to flexibly cope with and overcome these double uncertainties. The emotional labor of riders makes up for the loopholes and blind spots behind the OLPs’ algorithmic management in a human way, but these strategies can only alleviate the tension between the platform and riders in the sense of representation, and also cause many psychological problems for riders. Therefore, we selected the take-out platform and take-out riders as the research object to study the influence of OLPs’ algorithmic management on the psychological health of workers.

In this paper, we conducted a qualitative study of more than three hundred take-out riders in Qingdao, China, which was supplemented by interviews with take-out platforms executives and engineers, to address the question: “*how does OLPs algorithmic management which is implemented by the take-out platforms affect psychological health of platform workers”?* We used grounded theory [11,12] to analyze the collected empirical fact data, and abstracted the theory on the basis of empirical facts, so as to comprehensively and objectively reveal the correlation between OLPs’ algorithmic management and workers’ psychological health factors.

## 2. Related Works

At the early development stage of OLPs, researchers generally believed that it was an effective mode for enterprises to achieve flexible employment and workers to achieve independent employment. Sloboda [13] believes that platform workers can flexibly and autonomously choose when, where, and how much work to do, and they can also choose to work on multiple platforms at the same time to reduce their dependence on any one platform. Thomas [14] also found that workers could freely arrange working hours, based on the business operation architecture designed by programs and customer information resources, without being controlled and constrained by employers. However, with the exposure of many problems of OLPs and the continuous progress of related studies, researchers gradually found that this so-called flexibility and autonomy are closely related to the uncertainty of various factors. Stewart et al. [15] believe that the work autonomy promised by flexible employment arrangements is unrealistic, and the work autonomy of OLPs is also affected by algorithm control. In the process of studying crowdsourcing workers, Schoerpf et al. [16] found that the platform had an important impact on the working hours, income, and creativity of crowdsourcing workers, as well as their working and living conditions, by using online evaluation mechanisms and other algorithms. Through the research on take-out riders, Ingrao [17] found food delivery workers’ job characteristics and how they can change these job characteristics to improve levels of well-being via job crafting in Italy. Duggan et al. [18] pointed out that the OLPs would manage and supervise the labor process in which workers participate in production and create value, and their autonomy in working time and task arrangement would be reduced due to the influence of intelligent algorithm control system.

Algorithmic management had emerged in OLPs as a method of organizing and coordinating extremely large groups of workers and clients in an automated way. The existing research mainly explored the role of algorithmic management in OLPs from two aspects: how to realize the rapid matching between supply and demand of the online labor market and how to implement and innovate the traditional human resource management process (including task allocation, behavior control, and performance evaluation). Wood et al. [19] believed that algorithmic management realized the rapid and accurate matching of labor supply and demand, and OLPs could quickly complete the accurate matching of labor and task demands by virtue of its technical information advantages. This efficient value creation method under the on-demand economy provided more opportunities for workers to work flexibly and independently. Idowu et al. [20] concluded, when studying the digital labor market, that algorithmic management has weakened the demand for human resource managers, and online platforms that use algorithmic technology as a virtual automation management role can save a lot of marginal costs and labor costs. Rani et al. [21] believe that the increasingly detailed division of labor led to the gradual quantification of work task content, which provided an opportunity for workers on OLPs to automatically assign and evaluate work tasks through algorithms.

Based on the above researches and analysis, it can be found that OLPs built a supply and demand matching transaction platform through digital technology, which allows customers to release task demands independently and workers to obtain flexible and independent employment. They adopt big data-driven algorithmic management methods to build a control system, so as to avoid uncertainty risks and ensure the quality of business operations. But the algorithmic management of OLPs also has a huge influence on the work autonomy of workers. Grabher et al. [22] found that Uber designed a set of automatic matching algorithms to ensure the high scheduling rates of online resources by limiting the rejection rate of drivers, and the intelligent algorithm would give priority to those drivers with high comprehensive ratings, thus reducing the autonomy of workers in obtaining jobs from a technical perspective. Matherne et al. [23] found that the Uber platform often pushed information to drivers through an algorithm system to remind drivers to improve their service attitude and work quality in order to control workers’ deviant and uncivilized behaviors. Jadhav et al. [24] found, in their research on take-out platform riders, that the platform not only grasped information sources and riders’ data, but also conducted real-time dynamic monitoring based on riders’ personal characteristics. The intelligent voice assistant developed by platforms, instead of a manual one, could constrain and control riders.

The existing OLPs’ algorithmic management research focused more on the influence mechanism on work autonomy and the influence on labor process. However, the implementation of algorithmic management on the online labor platform has actually constructed a work situation with higher work requirements and pressure [25,26], and there are few studies on how workers can independently choose to continuously extend working hours and improve labor intensity, thereby affecting the physical and mental health of workers. Shevchuk et al. [27] pointed out that in response to the “task crisis” brought on by OLPs, workers usually work hard to stay online 24 h a day, which greatly increased the labor reinforcement from mental level and behavioral levels. Those workers with poor economic conditions who need to bear various family responsibilities will independently extend their working hours and also bear various risks. This paper will quantitatively analyze how OLPs’ algorithmic management has a specific impact on the psychological health of workers through labor reinforcement from mental and behavioral levels.

## 3. Materials and Methods

Proceeding from the labor process theory and combining the research status of the gig economy, we conducted an intensive study of a single case with the purpose of generalizing from description to theory. In this paper, an online take-out platform (Meituan) and its dynamic relationship with take-out riders was taken as observed unit. Meituan is the largest takeout platform in China. According to the data released on the Meituan official website, in 2021, the Meituan takeout platform will have 250 million customers, 5.27 million active takeout riders, covering more than 2800 cities, with a transaction amount of 702.1 billion yuan and a total of 14.4 billion orders.

Our case selection was guided by the extreme case selection method [28], which is particularly useful for constructing new theory. The online take-out platform scores highly on both algorithmic matching and algorithmic control, which make it a particularly useful case from which to construct new theory on OLPs’ algorithmic management influence on psychological health of workers on OLPs.

### 3.1. Data Collection and Analysis

In process of data collection, multiple sources of qualitative data, including platform public data, written dialogue, and psychological interviews were used. All data, except those drawn from platform public data, were anonymized to ensure the privacy.

The data collection of our research is the riders of Meituan takeout platform in Qingdao, China. Qingdao Federation of Trade Unions established labor unions for take-out riders and accelerated to attract them to join trade unions to the greatest extent. The author, as the vice chairman of Qingdao Federation of Trade Unions, is responsible for this work all the year round. Therefore, 220 Meituan riders, who all work at a take-out distribution station affiliated with the Meituan platform in Qingdao, were selected as the research objects in this study. Through improving the working mechanism of labor rights protection and providing mental health services through the trade union, we gradually contacted the takeout riders to understand their work and service process.

In the first stage of data collection, unstructured interview was adopted to encourage the take-out riders to express their personal views around the take-out theme. The content of the communication was converted into text storage, and, then, topics and questions with high repeatability were comprehensively extracted. At last, we interviewed the riders with specific questions until the interview outline was completed.

In the second stage of data collection, semi-structured interview was adopted, and the interview content was flexibly adjusted according to the outline and the take-out riders’ answers, until the 220th rider, which basically covered the needs of this study. By October 2022, 275 in-depth interviews had been conducted in this study, with an average of 30 min to 10 h per respondent. Meanwhile, EAP (Employee Assistance Program) mental health services were provided for 220 take-out riders at least twice each. The mental and emotional conditions of take-out riders were recorded through professional mental scales and mental counseling.

The period, method, role, focus, and purpose of data collection in each stage are given in Table 1.

The interviewees in this study include 220 take-out riders. The demographic characteristics of the interviewees are shown in Table 2.

From Table 2, it can be seen that the majority of riders were male, and the age was mainly between 21 and 30. This means that the OLPs attract a large number of young male workers. In addition, the qualifications of riders were generally low: some riders had worked in the manufacturing industry and traditional service industry for some time, especially during COVID-19, while some riders chose to join the take-out riders because of the closure of the original factory or the layoff of the company.

The job characteristics of the interviewees are shown in Table 3.

From Table 3, it can be seen that the income of take-out riders is attractive (the average annual income of interviewees last year was 72,327 yuan). However, their work intensity is also very high, and they raise their income based on the amount of delivery orders, which is obtained at the expense of physical labor and sleep time. According to the data analysis of interviews, most of the take-out riders worked longer than the legal 40 h per week, and nearly 94% of the take-out riders worked longer than 8 h per day. From 11 to 12 noon was their busiest time, and nearly 65% of the take-out riders worked seven days per week. On average, a rider needs to deliver 46 orders every day and travels nearly 170 km.

### 3.2. Coding of Grounded Theory

The grounded theory emphasizes the promotion of theory from data, and believes that only through in-depth analysis of data the theoretical framework could be gradually formed. The core of grounded theory is to transform complicated empirical data into theoretical expression through coding.

Open coding is the process of splitting materials, comparing the similarities and differences between keywords and sentences, and conceptualizing and categorizing topics and events [29]. Because the data in the materials are usually inaccurate and scattered, it is necessary to redefine the interview data, analyze the materials sentence by sentence, and generate three-level nodes (A1,…, An) around the labor process of the take-out platform. This paper used qualitative analysis software to obtain 37 open coding nodes, including normative constraints, punishment mechanism, appeal mechanism, safeguard mechanism, etc. Some open coding is shown in Table 4.

Open coding will decompose and interpret the data, and get the possible internal connection of different concepts. Axial coding is intended to divide appropriate categories according to the similarity conditions, context, and interaction strategies of the analysis facts, and explore the correlation between the initial concepts through research and discussion. First of all, the interview outline focused on the labor control of the take-out platform, from which many factors about the platform control system could be obtained. The most talked about topics were rules regulations and science technology, so, in this paper, delivery specification, compensation system, and algorithm technology were classified. Secondly, according to demographic statistical characteristics, this paper classified the interviewees’ educational experience, household registration, marital status, and career planning as group characteristic factors. According to relevant research theories [30,31], this paper classified the concepts of high psychological pressure and higher economic income as social adaptation after comprehensive proofreading of the initial concepts. Finally, the factors related to platform participants were classified into agent factors, merchant factors, and consumer factors. To sum up, this paper classified 37 free nodes obtained by open coding and identified 8 categories. Axial coding results are shown in Table 5.

Selective coding is intended to abstract and categorize more than two categories formed by axial coding again. Explaining the relationship between main categories and corresponding categories is the core of the whole model. All corresponding categories were closely related to the main category, which provided a reasonable explanation for the factors that affected the mental health of workers caused by the online labor platforms’ algorithmic management. Through comparison with relevant research, the axial coding was further refined into four main categories: management system, technical means, labor factor, and participant factor, which are shown in Table 6.

According to the grounded theory, we can determine the influence factors of riders’ labor process. Due to the application of algorithmic management in OLPs, the labor control of take-out riders revolves around the core factor of time. Therefore, the influence factors of riders’ labor process can be divided into objective factors (system control factor, algorithmic control factor, and participant factor) and subjective factors (labor factor), which are shown in Figure 1.

In order to accurately identify the influence of online labor platforms’ algorithmic management on psychological health of workers, regression analysis modeling was used to analyze the relationship between influence factors and psychological health. Firstly, the psychological health of OLP workers was selected as the dependent variable, and, then, eight variables were selected from management system, technical means, labor factor, and participant factor. Variable selection and symbol description are shown in Table 7.

The regression analysis formula is defined as follows:$$y=\beta_{0}+\beta_{1}x_{1}+\beta_{2}x_{2}+\beta_{3}x_{3}+\beta_{4}x_{4}+\beta_{5}x_{5}+\beta_{6}x_{6}+\beta_{7}x_{7}+\beta_{8}x_{8}+\varepsilon$$
where $\beta_{0},\beta_{1},\beta_{2}\ldots$ are coefficients, and ε represents error.

Because the unit of selected variables are different, and the dimension difference of units is large, in order to eliminate the influence of dimension on regression, it is necessary to convert the data into standardized data with mean values of 0 and standard deviations of 1, and then the standardized multiple linear regression coefficient is obtained through regression. The regression results are shown in Table 8.

From Table 8, we can see that the adjusted R-square of the model was 0.9637, but the results of most parameter significance tests were not ideal. It shows that the model has some problems. First, is the question of multicollinearity should be considered. So, we needed to further calculate the variance inflation factor. Calculation result of variance inflation factor is shown in Table 9.

Generally, when the variance inflation factor of variables is greater than 100, it is considered that there is serious multicollinearity between them. From Table 9, it can be clearly observed that x1, x6, x7 and x8 had multicollinearity.

The regression results of the optimized model after removing the variable x6, x7, x8 are shown in Table 10.

It can be seen from Table 10 that, when the model contained x1, x2, x3, x4, x5, and intercept terms, the adjusted R-square was 0.9827, and the fit goodness of the model was the best. The parameters in the optimized model are significant. From the last column in the Table 10, we can see that the multicollinearity had been improved.

The results prove that delivery specification, compensation system, algorithm technology, group characteristic factors, and social adaptation, which are brought on by OLPs’ algorithmic management, directly affected the psychological health of workers.

### 3.3. Analysis of Psychological Scale

Perceived social support [32] refers to the social support that individuals can perceive subjectively, including the emotional feeling and satisfaction of being understood, respected, and supported that individuals perceive. Social adaptation can be reflected by perceived social support. Job burnout [33] refers to a state of exhaustion caused by long time, high intensity and high load of work. Its cause is that workers have high expectations of their work, resulting in bad moods, cognition effects, and other aspects. Delivery specification, compensation system, and algorithm technology can be reflected by job burnout. Psychological capital [34] is an individual’s understanding of self and a positive psychological state that an individual shows in the process of growth, including self-efficacy, hope, optimism, and resilience. Group characteristic factors can be reflected by psychological capital.

The positive effect of perceived social support on mental health has been confirmed by many studies, and the level of mental health can be further improved by improving perceived social support. Therefore, the hypothesis is proposed that perceived social support can significantly predict the mental health of take-out riders. The influence of perceived social support on mental health is expected to be positive, while the influence of job burnout on mental health is expected to be negative. Some studies show that, the more serious job burnout is, the worse the mental health is. Therefore, the higher the level of perceived social support of individuals, the higher the level of positive psychology, and the lower the possibility of job burnout. The hypothesis is proposed that job burnout plays an intermediary role in perceived social support and mental health. Previous studies have proved that there is a significant positive correlation between perceived social support and psychological capital. The higher the level of perceived social support, the more inclined people are to treat others with an optimistic and positive attitude, be good at building good interpersonal relationships, and show a more positive and healthy psychological state. Therefore, the hypothesis is proposed that psychological capital plays an intermediary role in perceived social support and mental health. Therefore, starting from perceived social support, job burnout, and psychological capital, this paper studies the influence factors of algorithmic management on the mental health of take-out riders.

In this study, 220 take-out riders were investigated with the Perceived Social Support Scale (PSSS), Maslach Burnout Inventory General Survey (MBI-GS), Psychological Capital Scale (PCS), and General Health Questionnaire (GHQ-12), and 52 riders were provided with mental counseling. The research carried out descriptive statistical analysis and correlation analysis on the scores of each scale.

According to previous research [35,36,37] on the relationship between perceived social support, job burnout, psychological capital, and mental health, the following research hypotheses were proposed:

**H1:** 
*Perceived social support can significantly predict the mental health of the take-out riders.*


**H2:** 
*Job burnout plays a partial intermediary role in perceived social support and mental health.*


**H3:** 
*Psychological capital plays a partial intermediary role in perceived social support and mental health.*


**H4:** 
*Job burnout and psychological capital play a chain intermediary role in perceived social support and mental health.*


By using the data of theses scales to conduct descriptive statistical analysis on riders’ perceived social support, job burnout, psychological capital, and mental health, the analysis results showed that the average score of perceived social support was (56.28 ± 16.26), the average score of job burnout was (32.79 ± 12.07), the average score of psychological capital was (92.36 ± 19.53), and the average score of mental health was (39.71 ± 8.06). The descriptive statistical results of research variables are shown in Table 11.

The independent sample *t*-test results of different gender riders on each variable are shown in Table 12.

As shown in Table 12, there were significant differences in perceived social support, psychological capital, and mental health variables between different genders, but there was no significant difference in job burnout.

Variance analysis on various variables of take-out riders with different daily working hours is shown in Table 13.

As shown in Table 13, there were significant differences in the scores of riders with different daily working hours for perceived social support, job burnout, psychological capital, and mental health variables.

Variance analysis on various variables of take-out riders with different daily delivery order quantity is shown in Table 14.

As shown in Table 14, there were significant differences in the scores of riders with different daily delivery order quantities for perceived social support, psychological capital, and mental health variables. However, there was no significant difference in job burnout.

Variance analysis of various variables of take-out riders with different monthly income is shown in Table 15.

As shown in Table 15, there are significant differences in the scores of riders with different monthly income for perceived social support, job burnout, psychological capital, and mental health variables.

The correlation analysis of perceived social support, job burnout, psychological capital, and mental health showed that there were correlations between each pairwise variable (*p* < 0.01). There was a significant negative correlation between perceived social support and job burnout, and the correlation coefficient r was between −0.139 and −0.336 (*p* < 0.01). There was a significant positive correlation between perceived social support and psychological capital, and the correlation coefficient r was between 0.436 and 0.582 (*p* < 0.01). There was a significant positive correlation between perceived social support and mental health, and the correlation coefficient r was between 0.337 and 0.497 (*p* < 0.01). There was a significant negative correlation between job burnout and psychological capital, and the correlation coefficient r was between −0.117 and −0.539 (*p* < 0.01). There was a significant negative correlation between job burnout and mental health, and the correlation coefficient r was between −0.471 and −0.512 (*p* < 0.01). There was a significant positive correlation between psychological capital and mental health, and the correlation coefficient r was between 0.468 and 0.668 (*p* < 0.01). Based on the proposed hypothesis and the analysis of the relationship between various variables, the intermediary effect between perceived social support, job burnout, psychological capital, and mental health was investigated. The Bootstrap method [38] was used to repeat sampling for 1000 times, and the confidence level was set at 95% to test the significance of these paths, as shown in Table 16.

The estimated value interval of all paths did not include 0. Perceived social support could directly predict the mental health of take-out riders, with a direct effect of 0.195. It could also mediate mental health through job burnout and psychological capital, and job burnout and psychological capital played a chain intermediary role between perceived social support and mental health of take-out riders, with a total indirect effect of 0.373.

## 4. Discussion

The management of large groups of highly independent, highly skilled workers on OLPs has been achieved through a variety of algorithms, which haveacted as a means of coordinating and controlling workers. Algorithmic control [39] refers to the use of algorithms to monitor platform workers’ behavior and ensure their alignment with the platform organization’s goals. Uber quantifies the driver’s work habits by recording all details of the driver’s whereabouts, thereby supervising the driver’s work process and improving the driver’s service quality. Although Uber has repeatedly promoted so-called hands-off management to give drivers full freedom and autonomy, it is implementing a higher level of monitoring, because it records a series of personal data of the driver. Shestakofsky [40] pointed out that algorithms in Uber give the company vast leverage over work processes and the mental health of drivers.

From the labor process of take-out riders’ perspective, we found that algorithmic management seriously affected the mental health of riders. In the rider’s whole labor process, the take-out platform is responsible for directing the rider’s work, the consumer is responsible for evaluating the rider’s work, and the take-out platform completes the final reward and punishment for the riders. The take-out platform can allocate orders to riders in a short time, calculate the estimated delivery time, plan the delivery route, while, at the same time, riders also face the arbitrary supervision of algorithm management. Any rider who fails to deliver the goods within the specified delivery time may be punished, a certain amount of money will be deducted from the rider once the delivery is overdue, and the rider cannot click on the delivery in advance. The take-out platform will determine whether the rider has violated the operation according to the location of the rider and the customer feedback mechanism. Once the rider clicks on the delivery in advance without the customer’s consent, a certain amount of money will be deducted. In addition, bad comments from customers are also one of the important reasons for the rider’s money deduction. When the harsh punishment system is used as the management basis and workers sell their emotions to the enterprise as part of the labor force, the workers need to face the dual emotional control of the platform and customers. OLPs transfer part of the labor supervision right to consumers. On the surface, the evaluation score is given by customers according to their service experience, but in fact, it is given by the platform through the algorithm scoring system. Riders rely on the dispatch mode, which is set by the platform algorithmic management to obtain remuneration. They usually choose to extend working hours and improve distribution intensity to increase income. From the above analysis, we can see that algorithmic management had a direct influence on the rider’s labor process, labor experience, and mental health.

According to the research and descriptive statistical analysis results, we can obviously conclude that there is a significant correlation between perceived social support, job burnout, psychological capital, and mental health. Among them there was a significant positive correlation between perceived social support and mental health, and perceived social support could positively predict the mental health of the take-out riders. Research hypothesis H1 is confirmed. From the perspective of different average daily working hours and daily delivery order quantity, the longer the working hours or the more the orders were, the lower the riders’ perceived social support, psychological capital, and mental health were, which is consistent with previous research results [41]. For the take-out riders, constructing a good interpersonal relationship, getting the understanding and support of family and friends, as well as social respect and recognition, is a process of strengthening social support, which will involve emotional experience and other issues, which will ultimately have a relationship with their mental health.

The social support that individuals feel can help them cope with stressful events and relieve emotions, which is conducive to mental health. The level of job burnout of the take-out riders who work more than 12 h a day was the highest, which may be due to the long working hours and high work intensity, because workload and time distribution are not only the influencing factors of job burnout, but also affect the enthusiasm of work and the sense of success in work. At the same time, the platform algorithmic management results in take-out riders working long hours and receiving more orders per day, which makes riders spend less time communicating with their family and friends. When there are fewer channels to obtain emotional communication from the outside, the level of perceived social support of individuals will also be reduced. The analysis results from different monthly income levels showed that the higher the monthly income was, the higher the perceived social support. Some studies have shown that the income gap may cause individuals with poor economic conditions to deviate in self-cognition, thus resulting in a reduction in subjective support in social support, namely, a reduction in perceived social support. In terms of job burnout, the level of job burnout of riders with incomes of 5000–7000 was lower than that of riders with monthly incomes of 5000 and below. It may be because the riders with higher monthly income need more working hours or order quantity, which easily leads to physical fatigue, and physical fatigue also easily leads to psychological fatigue. The highest level of job burnout of riders with monthly incomes of more than 10,000 indicated that high income makes job satisfaction higher and work enthusiasm higher, so it is not as easy to be tired, and further makes their psychological capital and mental health higher. Some studies [42,43] have found that, if individuals feel more supportive resources, such as organizational support or family support, it will help prevent job burnout. Therefore, perceived social support can negatively predict job burnout. If workers are stuck in job burnout for a long time, they will inevitably have an impact on their mental health. It can be seen that perceived social support not only had a direct impact on mental health, but also indirectly affected mental health through the partial intermediary role of job burnout. Research hypothesis H2 is confirmed.

Through interviews and research, it was found that most of the take-out riders were able to look optimistically at many problems encountered in their work (complaints, weather, etc.), and they believed that they could solve the problems themselves. On the whole, the psychological capital level of take-out riders was high. The level of psychological capital and mental health of riders with incomes above 10,000 were lower than those of riders with other monthly incomes, respectively. This result is different from the existing research results [44], wherein the higher the family income, the healthier the mental health. The specific reason for this result may be related to the relative satisfaction of work input and work income. Descriptive statistical analysis results show that perceived social support can positively predict psychological capital level. Therefore, perceived social support can indirectly affect mental health through psychological capital. That is, psychological capital played a partial intermediary role in perceived social support and mental health. Research hypothesis H3 is confirmed.

According to the above relationships between perceived social support, job burnout, psychological capital, and mental health, it was shown that perceived social support can affect the job burnout of the take-out riders, and job burnout has an impact on psychological capital, thus further affecting mental health. They proved that job burnout and psychological capital played a chain intermediary role in perceived social support and mental health. Research hypothesis H4 is confirmed.

Prolonging working hours and increasing labor intensity for a long time will not only cause physical damage, but also lead to mental health problems. Many studies [45,46] have shown that labor reinforcement can significantly reduce employees’ job satisfaction and happiness, and then have a negative influence on their psychology and behavior. According to the results of psychological evaluation and analysis in this paper, the mental health problems of riders were caused by the reduction of workers’ autonomy, the continuous occupation of life time by work tasks and the increasing workload caused by algorithmic management. Ogbonna et al. [47] pointed out that, with the increase in labor intensity, the energy and physical strength consumed by many consumptive work tasks needs to be recovered through rest, and the continuous consumption and inadequate recovery would lead to the gradual exhaustion of individual energy, which will have a negative influence on workers’ mood. Those findings are consistent with the conclusion of this paper. In addition, through psychological evaluation and analysis, it was found that the work identity of the rider was inconsistent with the current environment, which would lead to a sense of tension. This sense of tension caused by role conflict and cognitive ambiguity often leads to work stress and job burnout. The rider and other workers engaged in OLPs would put themselves in a high-risk and uncertain environment, and they were tired of running and being strictly controlled by the platform algorithm, which would make them feel confused about their own identity, which would then unconsciously adapt to their own identity. It shows that workers’ independent extension of working hours and improvement of labor intensity under the control of their autonomy would reduce their satisfaction and happiness, increase emotional exhaustion, and reduce the level of work performance.

## 5. Conclusions

The rise of the gig economy platform largely depends on internet technology and new organizational management models. This paper studied the labor process of take-out platforms from the perspectives of algorithmic management. It tried to find the answer to how online labor platforms’ algorithmic management affected the psychological health of workers. In this paper, the quantitative analysis method was used to analyze the various data, and the grounded theory method was used to find the main factors affecting the labor process and psychology of take-out riders. This study found that the mental health of the take-out riders was generally good, and there was less job burnout. However, there were significant differences in working hours, daily delivery order quantity, and monthly income; these factors directly affected the psychological health of take-out riders. This study deeply analyzed the influence of an online labor platform’s algorithmic management on work autonomy and the threat to workers’ physical and mental health. It provides a reference for standardizing the employment of platform enterprises to promote the sustainable development of the gig economy. In the future, I will define the concept of algorithmic management in a structured and systematic way, and I will take various methods to further promote empirical research.

## Figures and Tables

**Figure 1 ijerph-20-04519-f001:**
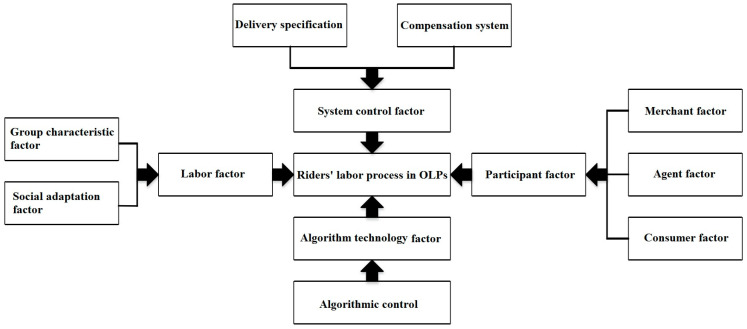
The influence factors of riders’ labor process and mental health.

**Table 1 ijerph-20-04519-t001:** Data collection in each stage.

Period	Sources/Methods	Purpose
Stage 1: June 2021–December 2021	● Platform public data: Order, meal delivery time, salary, and other data of riders in OLPs. ● Unstructured interview: Communicate with the take-out riders face to face around the take-out theme and encourage them to express their personal views.	● Understanding algorithmic management and its impact on platform work and riders. ● Open coding in grounded theory method.
Stage 2: January 2022–October 2022	● Structured interview: Riders’ feelings, emotions, and responses to control by the platform.	● Determining the influence factors of online take-out platform algorithmic management and riders’ labor process. ● Axial coding and selective coding in grounded theory method.
Mental health services: January 2022–October 2022	● Mental scales: Through emotional labor scale, psychology well-being scale, etc., obtaining various tensions experienced by riders in their everyday work. ● Mental counseling: Through psychological interviews, the pressure and anxiety of riders at work can be learned.	● Identifying the key factors for online take-out platform algorithmic management influence on psychological health of riders.

**Table 2 ijerph-20-04519-t002:** Demographic characteristics of interviewees.

Statistical Items	Values	Percentage
Gender	Male	85.9%
	Female	14.1%
Age	≤20	20.5%
	21–30	46.8%
	31–40	23.2%
	>40	9.5%
Qualifications	High school or below	68.6%
	Junior college	23.6%
	Undergraduate	7.8%
	Postgraduate	0%
Marital and Fertility state	Unmarried and childless	26.4%
	Married and childless	18.6%
	Married and childbearing	55%
Working years	Less than 1 year	29.1%
	1–2 years	26.4%
	2–3 years	22.7%
	More than 3 years	21.8%

**Table 3 ijerph-20-04519-t003:** Job characteristics of interviewees.

Statistical Items	Values	Percentage
Daily working hours	≤8 h	6.4%
	8–10 h	44.5%
	10–12 h	27.3%
	>12 h	21.8%
Working day of each week	<5	7.7%
	5–6	28.2%
	7	64.1%
Daily delivery order quantity	<30	12.3%
	30–40	27.7%
	40–60	41.8%
	>60	18.2%
Monthly income	<5000	20.9%
	5000–7000	34.6%
	7000–10,000	30.9%
	>10,000	13.6%
Average sleep time	<6 h	46.4%
	6–8 h	40.5%
	>8 h	13.1%

**Table 4 ijerph-20-04519-t004:** Open coding for interview materials.

Primary Coding	Reference Point	Example of Original Text
A1 Normative constraint	16	“There are safety education exams every month, and we have to read and do questions”
A2 Punishment mechanism	37	“The stationmaster said that it is not allowed to ask for leave for a long time. If no one can be found for three days, he will directly deduct money according to absenteeism”
A3 Appeal mechanism	15	“If the delivery is delayed because the customer does not answer the phone, the platform will not deduct money”
A4 Safeguard mechanism	8	“Insurance is only known to be deducted from wages. But it is not clear what insurance is and how much money is deducted”
A5 Delivery order mechanism	87	“If you can endure hardship, you can work more than ten hours a day and receive hundreds of orders a day”
A6 Rating mechanism	119	“If you can endure hardship, you can work more than ten hours a day and receive hundreds of orders a day”
A7 Encouragement mechanism	6	“Platform would give 666 yuan for 60 days of work”
A8 Dispatch mechanism	202	“15 orders should be delivered every day, so that the day can be counted as full attendance”
A9 Delivery time control	150	“The time limit for delivery is often very short. The distance of 6 km is only 15–20 min. Once the time limit is exceeded, you will be fined”
A18 Ownership of work	208	“I’m just a rider. I don’t have so many ambitions. I just want to live by my own work”
A26 Dispute resolution mechanism	166	“The customer asked me to walk 2 km to deliver the takeout to the new location. I didn’t agree. Later, the restaurant finally agreed to refund the bill and asked me to send the meal back. Finally, I got a subsidy of 3 yuan after working for two hours”

**Table 5 ijerph-20-04519-t005:** Axial coding for interview materials.

Category	Initial Concepts
B1 Delivery specification	A1 Normative constraint A2 Punishment mechanism A3 Appeal mechanism A4 Safeguard mechanism
B2 Compensation system	A5 Delivery order mechanism A6 Rating mechanism A7 Encouragement mechanism
B3 Algorithm technology	A8 Dispatch mechanism A9 Delivery time control
B4 Group characteristic factors	A10 Educational experience A11 Household registration A12 Marital status A13 Career planning
B5 Social adaptation	A14 High mobility A15 Low threshold for take-out industry A16 Short entry time A17 Working hours long A18 Ownership of work A19 Delivery timeout A20 Violation of traffic regulations A21 High psychological pressure A22 Feeling aggrieved A23 Low sense of identity A24 Work freedom A25 Higher income
B6 Agent factors	A26 Dispute resolution mechanism A27 Manager personality A28 Communication skills A29 Decision-making ability A30 Professional skills A31 Performance control A32 Price compensation mechanism
B7 Merchant factors	A33 Slow cooking speed A34 Bad attitude
B8 Consumer factors	A35 Additional service requirements A36 Malicious negative feedback A37 Poor communication attitude

**Table 6 ijerph-20-04519-t006:** Selective coding for interview materials.

Main Categories	Corresponding Categories	Specific Meaning of Category
C1 Management system	B1 Delivery specification	Work specifications and code of conduct for take-out riders
	B2 Compensation system	Salary system and incentive mechanism for take-out industry
C2 Technical means	B3 Algorithm technology	Control of platform labor process with the help of algorithm technology
C3 Labor factor	B4 Group characteristic factors	Demographic characteristics of workers in take-out industry
	B5 Social adaptation	Economic level, social interaction, and psychological identity level of take-out riders
C4 Participant factor	B6 Agent factors	Management style and attitude of the managers towards take-out riders
	B7 Merchant factors	Attitude of merchants towards take-out riders
	B8 Consumer factors	Factors of consumers affecting rider work

**Table 7 ijerph-20-04519-t007:** Variable selection and symbol description.

Variable Name	Symbol
Psychological health of OLP workers	y
Delivery specification	x1
Compensation system	x2
Algorithm technology	x3
Group characteristic factors	x4
Social adaptation	x5
Agent factors	x6
Merchant factors	x7
Consumer factors	x8

**Table 8 ijerph-20-04519-t008:** Regression results.

	Estimate	*t* Value	Pr (>|t|)
(Intercept)	−7.673 × 10^−16^	0.000	1.000
x1	2.451 × 10^−1^	1.177	0.3602
x2	6.998 × 10^−1^	2.985	0.0907
x3	−2.285 × 10^−16^	−0.147	0.8941
x4	1.033 × 10^0^	9.987	0.0089
x5	1.187 × 10^0^	10.052	0.0087
x6	1.127 × 10^−1^	0.093	0.9258
x7	−2.086 × 10^−1^	−0.152	0.8932
x8	4.388 × 10^−1^	0.163	0.8862

R-squared: 0.9912. Adjusted R-squared: 0.9637. F-statistic: 33.97. *p*-value: 0.0268.

**Table 9 ijerph-20-04519-t009:** Calculation result of variance inflation factor.

Symbol	Variance Inflation Factor
x1	130.22
x2	12.36
x3	13.51
x4	34.92
x5	75.22
x6	565.72
x7	469.29
x8	184.81

**Table 10 ijerph-20-04519-t010:** Regression results of optimized model.

	Estimate	*t* Value	Pr (>|t|)	vif
(Intercept)	−6.195 × 10^−16^	0.000	1.000	
x1	2.572 × 10^−1^	4.326	0.0079	1.89
x2	6.352 × 10^−1^	11.863	0.0091	1.56
x3	−2.529 × 10^−1^	0.019	0.0877	1.15
x4	1.155 × 10^0^	6.336	0.0081	1.83
x5	0.958 × 10^0^	8.629	0.0057	1.78

R-squared: 0.9907. Adjusted R-squared: 0.9827. F-statistic: 132.85. *p*-value: 2.581 × 10^−5^.

**Table 11 ijerph-20-04519-t011:** The descriptive statistical results of research variables.

Items	M	SD
Perceived social support	56.28	16.26
Job burnout	32.79	12.07
Psychological capital	92.36	19.53
Mental health	39.71	8.06

**Table 12 ijerph-20-04519-t012:** The independent sample *t*-test results of different gender riders.

Items	Male (n = 189)	Female (n = 31)	*t*
	M ± SD		
Perceived social support	56.92 ± 15.89	49.75 ± 16.06	2.93 **
Job burnout	32.85 ± 13.11	33.67 ± 9.96	−0.341
Psychological capital	99.98 ± 21.75	76.55 ± 20.73	6.87 ***
Mental health	39.96 ± 5.72	33.98 ± 6.81	5.98 ***

** represents *p* < 0.01, *** represents *p* < 0.001.

**Table 13 ijerph-20-04519-t013:** Variance analysis on various variables with different daily working hours.

Items	≤8 h	8–10 h	10–12 h	>12 h	*F*
	M ± SD				
Perceived social support	57.65 ± 13.92	57.78 ± 18.19	58.27 ± 15.27	47.62 ± 15.83	13.09 ***
Job burnout	31.63 ± 12.75	33.02 ± 11.86	35.79 ± 12.59	37.28 ± 9.56	4.72 **
Psychological capital	101.62 ± 20.38	99.71 ± 23.89	98.29 ± 23.61	86.11 ± 19.37	26.52 ***
Mental health	40.86 ± 6.12	40.26 ± 7.34	39.63 ± 7.52	36.14 ± 6.49	31.94 ***

** represents *p* < 0.01, *** represents *p* < 0.001.

**Table 14 ijerph-20-04519-t014:** Variance analysis on various variables with different daily delivery order quantity.

Items	<30	30–40	40–60	>60	*F*
	M ± SD				
Perceived social support	59.47 ± 14.85	56.25 ± 14.31	52.61 ± 16.34	48.03 ± 17.16	12.78 ***
Job burnout	32.35 ± 12.71	32.32 ± 12.37	34.51 ± 12.16	36.56 ± 10.27	2.49
Psychological capital	102.35 ± 21.06	97.52 ± 24.04	92.82 ± 21.36	78.83 ± 21.26	16.97 ***
Mental health	40.39 ± 6.09	39.51 ± 7.59	37.99 ± 7.49	34.28 ± 6.92	18.27 ***

*** represents *p* < 0.001.

**Table 15 ijerph-20-04519-t015:** Variance analysis on various variables with different monthly income.

Items	<5000	5000–7000	7000–10,000	>10,000	*F*
	M ± SD				
Perceived social support	57.32 ± 15.63	60.38 ± 14.36	52.97 ± 18.23	47.11 ± 16.46	10.82 ***
Job burnout	33.85 ± 11.31	32.05 ± 13.26	32.61 ± 12.12	34.86 ± 9.14	2.85 *
Psychological capital	92.81 ± 21.77	103.82 ± 20.14	97.91 ± 26.32	81.29 ± 22.88	16.59 ***
Mental health	39.06 ± 7.12	41.72 ± 5.93	40.36 ± 6.52	34.27 ± 8.83	16.14 ***

* represents *p* < 0.05, *** represents *p* < 0.001.

**Table 16 ijerph-20-04519-t016:** Bootstrap analysis of path effect significance test.

Path	Standardized Effect Estimated Value	Relative Effect Value	95% Confidence Interval	
			Lower Limit	Upper Limit
Perceived social support→Mental health	0.195	33.97%	0.062	0.319
Perceived social support→Job burnout→Mental health	0.085	14.96%	−0.207	−0.072
Perceived social support→Psychological capital→Mental health	0.246	42.95%	0.063	0.138
Perceived social support→Job burnout→Psychological capital→Mental health	0.042	7.91%	0.282	0.485

## Data Availability

Not applicable.

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
