# Peer review of "Quantitative Analysis of Online Labor Platforms’ Algorithmic Management Influence on Psychological Health of Workers"

_ijerph, 2023, doi:10.3390/ijerph20054519_

Round 1

Reviewer 1 Report

The research method is relatively simple, plain statistical analysis could not identify the accurate the influence of online labor platforms algorithmic management on psychological health of workers. The article should build an econometric model to analyze their relations.

Author Response

Response to the comments of reviewer

Dear Editor,

Thank you very much for your letter and for the comments by the reviewers. These comments are very valuable and helpful for my paper.

I appreciate the careful, constructive, and generally favorable reviews given to our paper by the reviewers.

I believe that I have adequately addressed all the excellent advices and questions raised by reviewers. Furthermore, I checked the manuscript and made sure the submitted manuscript is correct.

Point 1: The research method is relatively simple, plain statistical analysis could not identify the accurate the influence of online labor platforms algorithmic management on psychological health of workers. The article should build an econometric model to analyze their relations. 

Response 1: Thanks for the comment. I had built an econometric model (regression analysis model), and optimized it in the end of Section 3.2. Then the accurate influence of online labor platforms algorithmic management on psychological health of workers can be identified.

Reviewer 2 Report

INTRODUCTION

There are no data and or references about the workers pressure under the OLP's algoritmic management mode. The authors only mention some statements like accident, pressure, etc but there is no supporting data and or references.  Please mention some supporting data and references that shows issues of psychological health of the OLP workers.

The introduction is too brief. The authors needs to point out the criticality of these research more detail with more  supporting data and references.

RELATED WORKS

1. Again, the authors (only) mention some statements about workers psychological issue but they will be just statements or opinions if there are no supporting data and or references, for example: "However, the implementation of algorithmic management on online labor platform has actually constructed a work situation with higher work requirements and pressure, and there are few  studies on how workers can independently choose to continuously extend working hours and improve labor intensity, thereby affecting the physical and mental health of workers."

2.  There are at least 4 factors investigated in this research : 1. OLPs algorithmic management, 2. the psychological health of workers, 3. labor reinforcement from mental level 4. labor reinforcement from behavioral levels. Only first and second variables as the first (input) indicator and last (output) indicator that are mentioned in the INTRODUCTION and RELATED WORKS (the second variables were explained more like assumption without supporting data and references). Then, how about the third and fourth variables? What are the specific reason? what are the supporting data and references about those remaining two variables?

MATERIALS AND METHODS

1. Are the 220 riders who were selected as the research object represents population of specific object or they are sample of more population? if so, what are the reason of choosing them as samples? what are the sampling methods and how it resulted to 220 riders?.

2. Data collection: What are the reason behind the period? why was stage one conducted much earlier than the stage two and why was "Mental health services' was conducted in the same period with stage two? what was meant by "mental health service" mentioned in the table?

3. Are the interview items selected from certain references? what are they?

4. Looking at the interviewee background, most of them are male. Was it represent the workers as the research object and did it support the problem solving? prove it!

5. The hypothesis mentioned needs to have some supporting references

6. According to the statistical result, are all hypothesis accepted? please give more systematic result related to the hypotheses. For example, how many hypotheses.. what is the code of each hypothesis like hypothesis 1, 2, etc and which hypothesis are accepted and rejected.. etc.

DISCUSSION

Please provide more systematic discussion. I think the focus of this paper is the hypotheses proposed. It would be better if the discussion are written in different paragraph in which each paragraph explained the discussion of each proposed hypotheses results.

CONCLUSION

I don't see any concluded results in the conclusion. please add some brief results in the conclusion.

Author Response

Response to the comments of reviewer

Dear Editor,

Thank you very much for your letter and for the comments by the reviewers. These comments are very valuable and helpful for my paper.

I appreciate the careful, constructive, and generally favorable reviews given to our paper by the reviewers.

I believe that I have adequately addressed all the excellent advices and questions raised by reviewers. Furthermore, I checked the manuscript and made sure the submitted manuscript is correct.

Point 1: There are no data and or references about the workers pressure under the OLP's algoritmic management mode. The authors only mention some statements like accident, pressure, etc but there is no supporting data and or references. Please mention some supporting data and references that shows issues of psychological health of the OLP workers.

Response 1: Thanks for the comment. I added relevant data and references that shows issues of psychological health of the OLP workers.

Point 2: The introduction is too brief. The authors need to point out the criticality of these research more detail with more supporting data and references.

Response 2: Thanks for the comment. I had improved Introduction section.
More supporting data and references are given to point out the criticality of the research.

Point 3: Again, the authors (only) mention some statements about workers psychological issue but they will be just statements or opinions if there are no supporting data and or references, for example: "However, the implementation of algorithmic management on online labor platform has actually constructed a work situation with higher work requirements and pressure, and there are few  studies on how workers can independently choose to continuously extend working hours and improve labor intensity, thereby affecting the physical and mental health of workers."

Response 3: Thanks for the comment. I added relevant references.

Point 4: There are at least 4 factors investigated in this research: 1. OLPs algorithmic management, 2. the psychological health of workers, 3. labor reinforcement from mental level 4. labor reinforcement from behavioral levels. Only first and second variables as the first (input) indicator and last (output) indicator that are mentioned in the INTRODUCTION and RELATED WORKS (the second variables were explained more like assumption without supporting data and references). Then, how about the third and fourth variables? What are the specific reason? what are the supporting data and references about those remaining two variables?

Response 4: Thanks for the comment. I added relevant references. labor reinforcement from mental level and labor reinforcement from behavioral levels are important factors (variables). I had added supporting data and references about them in the Introduction and Related works sections.

Point 5: Are the 220 riders who were selected as the research object represents population of specific object or they are sample of more population? if so, what are the reason of choosing them as samples? what are the sampling methods and how it resulted to 220 riders?.

Response 5: Thanks for the comment. The 220 Meituan riders all work at a take-out distribution site affiliated to the Meituan platform in Qingdao. Meituan platform has several take-out distribution stations in each city. Therefore, the works of each station are very representative. I also added relevant descriptions in the paper.

Point 6: Data collection: What are the reason behind the period? why was stage one conducted much earlier than the stage two and why was "Mental health services' was conducted in the same period with stage two? what was meant by "mental health service" mentioned in the table?

Response 6: Thanks for the comment. Because take-out riders are busy, they rarely receive unstructured interviews for a long time, and they can only use gaps to answer fragmented questions. Therefore, the first stage needs a long time to complete. "mental health service" mentioned in Table 1 refers to the psychological interview service provided by professional psychological counselling institutions for riders. In this process, riders completed some psychological scales and tests. Because of the busy work of riders, more than 200 riders cannot complete the psychological scale test in a short time, because the psychological service and structured interview can be conducted simultaneously.

Point 7: Are the interview items selected from certain references? what are they?

Response 7: Thanks for the comment. This study does not set up a fixed interview outline, but uses heuristic methods to constantly explore questions, from which we can learn about the daily work content, salary composition, recruitment information and relevant management systems of the take-out station and the take-out riders.

Point 8: Looking at the interviewee background, most of them are male. Was it represent the workers as the research object and did it support the problem solving? prove it!

Response 8: Thanks for the comment. Because take-out rider is a job with high physical requirements, the majority of workers are male. This paper (Sun P, Zhao Y, Zhang Q. Platform, gender and labor: gender performance of "female rider" [J] Essays on Women's Studies, 2021, 6: 5-16) also shows that most of take-out riders are male. In our research, the research object is not selected, but the whole workers of a take-out station. The gender ratio is naturally formed.

Point 9: The hypothesis mentioned needs to have some supporting references.

Response 9: Thanks for the comment. supporting references for the hypothesis had been added.

Point 10: According to the statistical result, are all hypothesis accepted? please give more systematic result related to the hypotheses. For example, how many hypotheses.. what is the code of each hypothesis like hypothesis 1, 2, etc and which hypothesis are accepted and rejected.. etc..

Response 10: Thanks for the comment. There are four hypotheses, and I added these hypotheses before Table 7 in the paper. The four hypothesis are all accepted.

Point 11: Please provide more systematic discussion. I think the focus of this paper is the hypotheses proposed. It would be better if the discussion are written in different paragraph in which each paragraph explained the discussion of each proposed hypotheses results.

Response 11: Thanks for the comment. The focus of this paper is indeed the hypotheses proposed. I had revised and improved Discussion section.

Point 12: I don't see any concluded results in the conclusion. please add some brief results in the conclusion.

Response 12: Thanks for the comment. I had revised and improved Conclusion section.

Round 2

Reviewer 1 Report

The paper has made a major revision and added the quantitative analysis model and research hypothesis of the labor psychology of selling riders. The whole analysis process is more reasonable and the research conclusion is more credible. But there are too many references in Chinese. It is suggested to supplement the literature on the labor psychology of takeaway riders from other countries except China.

Author Response

Response to the comments of reviewer

Dear Editor,

Thank you very much for your letter and for the comments by the reviewers. These comments are very valuable and helpful for my paper.

I appreciate the careful, constructive, and generally favorable reviews given to our paper by the reviewers.

I believe that I have adequately addressed all the excellent advices and questions raised by reviewers. Furthermore, I checked the manuscript and made sure the submitted manuscript is correct.

Point 1: The paper has made a major revision and added the quantitative analysis model and research hypothesis of the labor psychology of selling riders. The whole analysis process is more reasonable and the research conclusion is more credible. But there are too many references in Chinese. It is suggested to supplement the literature on the labor psychology of takeaway riders from other countries except China. 

Response 1: Thanks for the comment. I had added and replaced some literatures on the labor psychology of takeaway riders from other countries except China.
